# MULTI-BATCH REINFORCEMENT LEARNING VIA SAMPLE TRANSFER AND IMITATION LEARNING

## ABSTRACT

Reinforcement learning (RL), especially deep reinforcement learning, has achieved impressive performance on different control tasks. Unfortunately, most online reinforcement learning algorithms require a large number of interactions with the environment to learn a reliable control policy. This assumption of the availability of repeated interactions with the environment does not hold for many real-world applications due to safety concerns, the cost/inconvenience related to interactions, or the lack of an accurate simulator to enable effective sim2real training. As a consequence, there has been a surge in research addressing this issue, including batch reinforcement learning. Batch RL aims to learn a good control policy from a previously collected dataset. Most existing batch RL algorithms are designed for a single batch setting and assume that we have a large number of interaction samples in fixed data sets. These assumptions limit the use of batch RL algorithms in the real world. We use transfer learning to address this data efficiency challenge. This approach is evaluated on multiple continuous control tasks against several robust baselines. Compared with other batch RL algorithms, the methods described here can be used to deal with more general real-world scenarios.

## 1 INTRODUCTION

Reinforcement learning aims to learn an optimal control policy through interactions with the environment (Sutton & Barto, 2018). Deep reinforcement learning (Deep RL or DRL) combines neural networks with reinforcement learning and further enables RL agents to deal with more complex environments. DRL has achieved impressive successes in different areas including the game of Go (Silver et al., 2017), Atari games (Mnih et al., 2015), and continuous control tasks (Lillicrap et al., 2015). However, deploying RL algorithms for real-world problems can be very challenging. Compared with supervised learning algorithms (e.g., classification or regression), most reinforcement learning algorithms need to interact with the environment many times to learn a reliable control policy. This process can be very costly or even dangerous for some real-world applications, e.g., safety-critical applications. The current success of deep RL algorithms heavily depends on a large number of interactions with the environment. Thus the practical application of reinforcement learning algorithms in the real world is critically limited by its poor data efficiency and its inflexibility of learning in an offline fashion.

Batch reinforcement learning (also known as offline reinforcement learning) algorithms have been developed to solve this issue. Batch RL aims to learn a control policy from a previously collected dataset without further interactions with the environment. Batch reinforcement learning has attracted a considerable amount of attention due to its potential in dealing with real-world problems. There have been many efforts in developing batch RL methods: early approaches include fitted Q iteration method (Ernst et al., 2005) which uses a tree-based model to estimate the state-action value function (Q function) in an iterative way, the neural fitted Q method (Riedmiller, 2005), which leverages a multilayer perceptron (MLP) to approximate the Q function. Since these earlier efforts, a number of batch reinforcement learning algorithms have been developed that further improve the learning performance. These approaches can be generally categorized as Q function based methods (Fujimoto et al., 2019; Kumar et al., 2020) and imitation learning based methods (Wang et al., 2018; Peng et al.,

2019). Examples of Q function based methods include (Fujimoto et al., 2019), in which the authors constrain updates of Q values to deal with distribution drift, and (Kumar et al., 2020), where an additional penalty term was added to constrain the update for the Q function. Best Action Imitation Learning (BAIL) (Chen et al., 2019) , on the other hand, leveraging the so-called upper envelope, directly select a sub-batch of the dataset and then execute imitation learning on the selected data to learn the control policy.

Even with these advances, most existing batch RL algorithms assume that we have a large number of data points in the batch. In the real world, this may be an unrealistic assumption. For example, when learning energy management control policies, we may only have a very limited amount of collected data for newly built houses and buildings. It is hard for most current batch RL algorithms to learn a reliable policy with a limited amount of data points. In this work, we use transfer learning (Torrey & Shavlik, 2010) to address this issue. Transfer learning aims to use the knowledge from source domains (domains for which we have a large amount of data) to improve the learning performance in the target domain (the domain we are interested in, but for which we only have a limited amount of data). Depending on the manner in which knowledge is reused from the source domains, there are three main categories of transfer learning: sample transfer, model transfer, and representation transfer. In this work, we use sample transfer, i.e., we transfer some related data points from the source tasks to improve the learning performance in the target control task. The proposed algorithm is referred to as BAIL+ as it is a direct extension of the BAIL algorithm proposed by (Chen et al., 2019).

Multitask learning (MTL) (Caruana, 1997) aims to learn a set of tasks jointly instead of learning them separately to achieve a better overall learning performance. In general, multitask learning can help to improve the data efficiency via reusing the shared representations and training on data from multiple sources. Multitask learning has shown its effectiveness on different applications such as image classification (Li et al., 2015; 2018), nature language processing (Collobert & Weston, 2008; Liu et al., 2019), and speech recognition (Siohan & Rybach, 2015). It has shown to be useful in deep reinforcement learning field in recent years (Li et al., 2019). The main objective is to learn one (or more) policy that can perform well on multiple tasks. In (Rusu et al., 2015), the authors proposed to learn a multitask policy from a set of DQN experts that are pre-trained on a set of tasks separately. Lately, (Hessel et al., 2019) investigated how to balance the learning over multiple tasks with one deep neural network based policy network. The benefit of representation sharing is also investigated in (D'Eramo et al., 2019) for learning deep RL based control agent. Most of existing multitask RL works are based on online RL. There are some preliminary investigations on batch RL. As shown in (Li et al., 2019), the authors proposed to learn distill pretarined BCQ models into one model. However, despite a few attempts, how to further improve the task-level generalization for batch RL is still an open question. In this work, different from previous works, we propose to first improve the learning performance on single tasks and then utilize the policy distillation to combine the learned policies into one single policy.

Batch RL algorithms are typically designed to deal with single task scenario (a single batch setting). In the real world, it is more common to have batches collected from a set of tasks that have similar Markov Decision Process (MDP) settings. For example, we may have collected datasets from a set of houses/buildings in the same area. Thus, it will very helpful if one general policy that can be learned from different bacthes that perform well on these different tasks even including unseen tasks without further adaption. In (Li et al., 2019), the authors implemented preliminary investigations on multitask batch reinforcement learning. In our work, to improve the task-level generalization of the policy learned with batch RL, we further extend BAIL+ for multi-batch settings via policy distillation. The resulting algorithm is referred to as MBAIL. Specifically, we aims to learn a general policy without the need to inferring the task identity which can make the batch RL more applicable in real world.

The remainder of this paper is organized as follows. The preliminaries of this work including batch reinforcement learning, multi-batch reinforcement learning, and BAIL algorithm are presented in Section II. The details for BAIL+ and MBAIL are presented in Section III. The effectiveness of the proposed methods are showcased in Section IV. Finally, conclusions and future work are presented in Section V.

## 2 PRELIMINARIES

This section reviews the main technical background and notation used in this paper including reinforcement learning, batch reinforcement learning, BAIL, and multi-batch reinforcement learning.

**Reinforcement Learning**   Reinforcement learning (Sutton & Barto, 2018) is a learning paradigm in which a learning agent aims to learn an optimal control policy by interacting with the environment. Reinforcement learning has been successfully applied in many areas including transportation (Haydari & Yilmaz, 2020), smart grids (Yang et al., 2020), and recommendation systems (Zou et al., 2019). Formally, an RL problem is typically formulated as a Markov Decision Process (MDP), i.e., a tuple $\langle \mathcal{S}, \mathcal{A}, p, r, \mu, \gamma \rangle$, where $\mathcal{S}$ is the state space, $\mathcal{A}$ is the action space, $p : \mathcal{S} \otimes \mathcal{A} \to \mathcal{S}$ is the state transition function, $r : \mathcal{S} \otimes \mathcal{A} \to \mathbb{R}$ is the reward function, $\mu$ is the initial state distribution, and $\gamma$ is the discount factor. The solution to an RL problem (control policy) is a function: $\pi : \mathcal{S} \to \mathcal{A}$. To obtain this solution, the agent needs to achieve the maximum expected cumulative reward, that is, the so-called value function. Given a state $s \in \mathcal{S}$ and a policy $\pi$, the state value function (a.k.a., value function) under policy $\pi$ is defined by $V^\pi(s) = \mathbb{E}^\pi(R_1 + \gamma R_2 + \cdots \gamma^{n-1} R_n)$, where $R_i$ denotes the reward obtained at time step $i$.

**Batch Reinforcement Learning**   In batch reinforcement learning (batch RL), the goal is to learn a high performance control policy using an offline dataset without further interactions with the environment. The dataset consists of $N$ data points $\mathcal{B} = \{(s_t, a_t, r_t, s_t^{'})|t = 1, .., N\}$. In general, we have no prior requests on the policy used to collect the batch $\mathcal{B}$. $\mathcal{B}$ can be obtained while training an RL policy in a episodic fashion or running some other control policy (e.g., rule-based methods) in the same way. The distribution of states and actions in the dataset can be far from those induced by the current policy under consideration. This complicates computations related to evaluating and optimizing behaviors, and this has been the primary consideration of several Batch RL solution methods. To counter this extrapolation problem, (Fujimoto et al., 2019) proposed an algorithm to learn policies with soft constrain to lie near the batch, which alleviate the extrapolation problem.

**BAIL: Best Action Imitation Learning**   Best Action Imitation Learning (BAIL) (Chen et al., 2019) is a simple and computationally efficient imitation learning based batch RL algorithm. The core concept of BAIL is very simple: finding actions that can achieve high return for each state $s$ and then learning a control policy based on these selection state-action pairs. To be more specific, for a particular state-action pair $(s, a)$, let $G(s, a)$ denote the return starting in state $s$ and action $a$, under the policy $\pi$. Denote the optimal value function by $V^*(s)$. Then if the action $a^*$ satisfies $G(s, a^*) = V^*(s)$, $a^*$ is an optimal action for state $s$. The problem now becomes how to obtain $V^*$ in a batch setting. Since there is no further interaction with the environment, it is impossible to find $V^*$. Therefore we seek to eliminate as many useless state-action pairs in the batch as possible, to avoid the algorithm inferring bad actions. To do this, we estimate a supremum of the optimal value function $V^*$, which is referred to as the *upper envelope*. Given $\phi = (w, b)$, a neural network parameterized $V_\phi : \mathcal{S} \to \mathbb{R}$, a regularization weight $\lambda$ and a dataset $\mathcal{D}$ of size $m$, where $\mathcal{D}_i = (s_i, G_i)$ and $G_i$ is the accumulated return of the state $s_i$ computed within the given batch, then the upper envelope function $V^* : \mathcal{S} \to \mathbb{R}$ is estimated by minimizing the following loss function:

$$\min_\phi \sum_{i=1}^{m} [V_\phi(s_i) - G_i]^2 + \lambda ||w||^2 \qquad s.t.\ V_\phi(s_i) > G_i \quad \text{where } i = 1, 2, \cdots m \qquad (1)$$

Once the upper envelope function $V_\phi$ is estimated, the best state-action pairs can be selected from the batch data $\mathcal{B}$ based on the estimated $V_\phi$. One way of selecting such pair is that for a fixed $\beta > 0$, we choose all $(s_i, a_i)$ pairs from the batch data set $B$ such that:

$$G_i > \beta V_\phi(s_i) \qquad (2)$$

Typically, one can set $\beta$ such that $p\%$ of the data points are selected, where p is a hyper-parameter. In this work, we follow the same setting as (Chen et al., 2019), in which $\beta$ is set to ensure that approximately 25% of all the data points are selected for each batch.

---

**Algorithm 1** BAIL+: Best Action Imitation Learning with Multi-source Sample Transfer

---
**Input:** A target task batch $\mathcal{B}_t$ and $N$ source task batches $\mathcal{B}_1, \mathcal{B}_2, \cdots, \mathcal{B}_N$ and the pre-defined sample selection ratio threshold $\tilde{\alpha}$

  1:  Learn the upper envelope function $V_t$ and the reward function $\hat{r}_t$ for batch $\mathcal{B}_t$.
  2:  **for** $j = 1, \cdots, N$ **do**
  3:     **for** $d = 1, \cdots, M$ **do**
  4:        Denote the current state action pair by $(s_d^j, a_d^j)$.
  5:        Following equation 5, estimate return of sample $(s_d^j, a_d^j)$, denote by $\hat{G}_d^j$.
  6:        Compute the sample selection ratio $\alpha(s_d^j, a_d^j)$ via equation 4.
  7:        **if** $\alpha(s_d^j, a_d^j) > \tilde{\alpha}$ **then**
  8:           Append $(s_d^j, a_d^j)$ to dataset $\mathcal{B}_t$
  9:        **end if**
10:     **end for**
11:     Learn the final policy $\pi_t$ on $\mathcal{B}_t$ via imitation learning
12:  **end for**
**Output:** the final policy of the target task $\pi_t$

---

**Multi-Batch Reinforcement Learning**    To make batch RL more suitable for real-world applications, it is desirable that the learned control policy performs well in multiple situations. In this work, we aim to learn one RL agent from a set of batches sampled from a set of tasks $\{T_1, \cdots, T_N\}$. Then the multi-task (multi-batch) batch reinforcement learning can be formulated as:

$$\arg\max_{\theta} J(\theta) = \mathbb{E}_{T_i \sim p(T)}[J_{T_i}(\pi_\theta)] \tag{3}$$

where $J_{T_i}(\pi_\theta)$ is referred to as the performance of control policy $\pi_\theta$ on task $i$. Here $p(T)$ defines the task distribution and for each task $i$, we have a corresponding dataset consisting of K tuples $\mathcal{B} = \{(s_i^t, a_i^t, r_i^t, s_i^{t'})| t = 1, .., K\}$.

## 3 METHODOLOGY

In this work, we aim to improve the RL agent's performance over mutiple tasks given a set datasets collected from multiple task. We tackle this problem in two stages. We first improve the BAIL algorithm via sample transfer, which results in the BAIL+ algorithm, in which we have one target task with a set of related source tasks. Furthermore, leveraging policy distillation, we extend BAIL+ to MBAIL (**M**ulti-batch **B**est **A**ction based **I**mitation **l**earning with sample transfer) to achieve better generalization over multiple tasks. Below we will illustrate both of our methods in more details.

### 3.1 BAIL+

When dealing with a set of related tasks, it is natural to think about how to leverage data across all the tasks. To achieve this, one straight forward approach is through sample transfer. The core idea of sample transfer is to utilize samples from numerous source tasks to construct a comprehensive dataset to improve the learning on the target task. In terms of BAIL, we employ an effective approach, that is, given a target task $T_t$, a state action pair $(s, a)$ from any source task and its trajectory $\eta_{s,a} = ((s, a), (s_1, a_1), \cdots, (s_k, a_k))$, we define the sample selection ratio of the state action pair $(s, a)$ similar to the definition of BAIL's sample selection ratio $\beta$, i.e.

$$\alpha(s, a) = \frac{\hat{G}^t(\eta_{s,a})}{V^t(s)} \tag{4}$$

Note $\hat{G}^t(\eta)$ is the estimated return of the source task samples evaluated on the target task and the same is for $V^t(s)$. Then given a selection threshold $\tilde{\alpha}$, if any state action pair $(s, a)$ has $\alpha(s, a) > \tilde{\alpha}$, we will incorporate this pair into our newly selected batch. The motivation for this is that, by assuming the correct estimation of $G_t$ and $V^t$, we just need to follow the original BAIL routine to pick the state action pair that induces the best action.

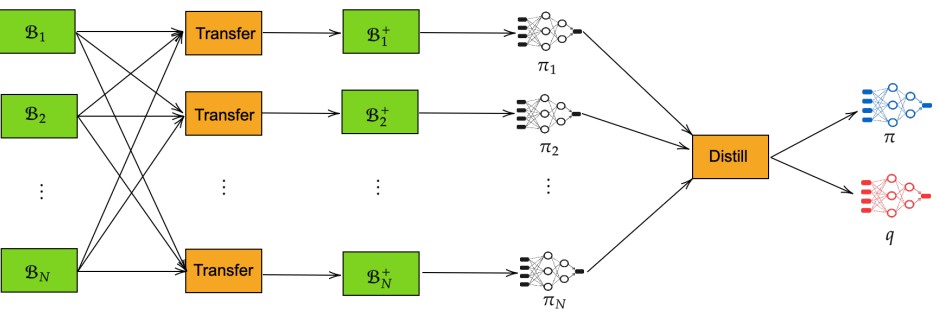

Figure 1: The overview for multi-batch BAIL.

Then, the imminent problem is to obtain an estimate of the return $\hat{G}^t$ evaluated on the target task. To solve this, we first learn a reward function on the target task $\hat{r}_t : \mathcal{S} \times \mathcal{A} \to \mathbb{R}$. Then given a discount factor $\gamma$ and a trajectory of state action pairs $\eta = (s_1, a_1), (s_2, a_2), \cdots, (s_M, a_M)$ from any source task batch, we can obtain its return estimate on the target task, that is:

$$\hat{G}^t(\eta) = \sum_{i=1}^{M} \gamma^{i-1} \hat{r}_t(s_i, a_i) \tag{5}$$

Note that we assume all tasks share the same transition function, and the batch is collected via the same policy, therefore Equation 5 is a reasonable estimation of the return on the target task. Once the return estimation is complete, we just need to select the samples based on the selection ratio function $\alpha$ and some threshold $\tilde{\alpha}$. Similarly to BAIL, we select $\tilde{\alpha}$ such that the top $p\%$ of all datasets from the source tasks. Once the data is selected, we then use a standard supervised learning based imitation learning method to obtain the final BAIL+ policy.

The pseudo code for BAIL+ is given in Algorithm 1. Assume we have one target task $T_t$ and $N$ source tasks $T_1, T_2, \cdots, T_N$ and their batch dataset $\mathcal{B}_t$ and $\mathcal{B}_1, \mathcal{B}_2, \cdots, \mathcal{B}_N$ of state action pairs. These tasks share the same state space, action space and transition functions, but have different reward functions. Assume $|\mathcal{B}_1| = \cdots = |\mathcal{B}_N| = M$. For the sake of simplicity, we also assume the length of all trajectories is $L$. Note this assumption can be easily lifted.

## 3.2 MULTI-BATCH BAIL+

In multitask reinforcement learning, we are often faced with a set of similar tasks and it is desirable to learn a policy that is able to leverage knowledge from all tasks and obtain a policy that has similar or better performance across all tasks. Policy distillation Rusu et al. (2015) is a classic multitask reinforcement learning approach, where the distillation agent aggregates knowledge from all of the policies and distill them into one consistent policy (Rusu et al., 2015). This distillation process leverages knowledge from all tasks and thus can potentially further improve policy performance. In this section, we will introduce our approach for multitask batch reinforcement learning based on the previous BAIL+ method, we will refer to this method as MBAIL.

Given a set of policies $\Pi = \{\pi_i | i = [1, 2, \cdots, N]\}$ and corresponding tasks $T_1, \cdots, T_N$, we want to learn a policy $\pi : \mathcal{S} \to \mathcal{A}$ such that $\sum_{i=1}^{N} \sum_{s \in \mathcal{B}_i} d(\pi(s), \pi_i(s))$ is minimized, where $d$ is a distance measure, and is chosen to be L2 distance. In addition, to help the task identification, we incorporate a task inference module $q : \mathcal{S} \times \mathcal{A} \times \mathbb{R} \times \mathcal{S} \to \mathbb{R}^k$. The distilled policy and the task inference module are all parameterized by a neural network. Denote the context tuple $c = (s, a, R, s')$, our algorithm aims to minimize the following loss function:

$$\mathcal{L}_\pi = \frac{1}{N} \sum_{i=1}^{N} \mathbb{E}_{s, c_i \sim \mathcal{B}_i} [(\pi_i(s) - \pi(s, z_i))^2 + \beta \mathrm{KL}(q(c_i) || \mathcal{N}(0, 1))], z_i \sim q(c_i) \tag{6}$$

However, this approach is not directly deployable under the batch setting. In (Li et al., 2019), the authors observed that the task inference module has learned to model the posterior over task identity as conditionally dependent on only the state-action pairs, but omit the effect of reward, which is

---

**Algorithm 2** MBAIL: Best Action Imitation Learning for Multiple Batches

---

**Input:** Batches $\mathcal{B}_1, \cdots, \mathcal{B}_N$ of $N$ tasks, maximum number of epochs $E$
1: **for** $t = 1, \cdots, N$ **do**
2:     Following Algorithm 1, train policy $\pi_t$.
3: **end for**
4: **for** $i = 1, \cdots, E$ **do**
5:     Compute the distillation loss $\mathcal{L}_\pi$ via Equation 6
6:     Compute the triplet loss $\mathcal{L}^{triplet}$ via Equation 7.
7:     Do gradient descent w.r.t. $\pi$ and $q$ for the loss function: $\mathcal{L} = \mathcal{L}^{triplet} + \mathcal{L}_\pi$
8: **end for**
**Output:** The distilled policy $\pi$ and the task inference module $q$.

---

crucial in the multitask setting. The reason for this behavior lies in the fact that there is no overlap (or little) between each batch. Therefore in this case, minimizing Equation 6 only leads to the algorithm learning the trivial correlations.

To avoid this problem, in (Li et al., 2019), the authors propose to add an additional loss function, namely the *triplet loss*. The motivation behind this loss is to enforce reward information to take part in the task inference. The authors achieve this by introducing a *relabeling* process. Given a context tuple $c_i = (s_i, a_i, R_i, s_i^{'})$ from batch $\mathcal{B}_i$ and a reward estimation of task $j$, $\hat{r}_j : \mathcal{S} \times \mathcal{A} \rightarrow \mathbb{R}$, the relabelling of $c_i$ to task j, denoted by $c_i^j$, is defined as: $c_i^j = (s_i, a_i, \hat{r}_j(s_i, a_i), s_i^{'})$. Then the triplet loss function is defined by:

$$\mathcal{L}^{triplet} = \frac{1}{N(N-1)} \sum_{i=1}^{N} \sum_{j=1, j \neq i}^{N} [d(q(c_i^j, q(c_i))) - d(q(c_i^j), q(c_j)) + a]_+ \qquad (7)$$

where $a$ is the triplet margin, $[\cdot]_+$ is the ReLU function , $q$ outputs the posterior over task identity and $d$ is a divergence measure which is chosen to be the KL diverge. Through minimizing Equation 7, we essentially encourage $q$ to infer similar task representations when given either $c_i$ or $c_i^j$. Moreover, it helps enforcing $q$ to infer different task identities for $c_i^j$ and $c_j$, which forces $q$ to account for the reward information instead of only relying on the state-action pairs.

Now, to add with the previous distillation loss, our multi-batch policy distillation loss is:

$$\mathcal{L} = \mathcal{L}^{triplet} + \mathcal{L}_\pi \qquad (8)$$

By minimizing the loss function, we will be able to obtain the final distilled policy $\pi$, as well as the inference module $q$. The pseudo-code of MBAIL is summarized in Algorithm 2. There are two main stages for MBAIL. In the first stage, BAIL+ is used to train policies for each task or each group of tasks with identical properties for BAIL+. In the second stage, the policies learned from each of the individual policies are distilled into one single multitask policy.

## 4 EXPERIMENTS

In the experiment, we evaluate both BAIL+ and MBAIL on a set of datasets introduced in (Li et al., 2019). As baseline comparisons, we have also included the results for single task BAIL, as well as contextual BCQ, where the networks in BCQ are modified to accept the inferred task identity as input as illustrated in (Li et al., 2019).

### 4.1 DATASET DESCRIPTION

The datasets consist three challenging task distributions, namely Ant-Dir, Ant-Goal and HalfCheetahVel, from MuJoCo (Todorov et al., 2012). For the multi-task setup, we utilize the exact data and evaluation protocols from (Li et al., 2019). In Ant-Dir, a target direction defines a task, where the agent maximizes returns by running with maximal speed in the target direction. In Ant-Goal, a task is defined by a goal location, to which the agent should navigate. In HalfCheetahVel, a task is defined as a constant velocity the agent should achieve. As specified by the dataset authors (Li et al.,

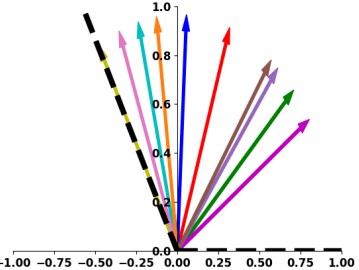

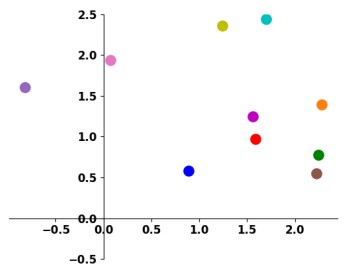

(a) The directions of all 10 tasks for Ant-Dir. The dashed black line indicates the angle border of the task distribution. The simulated robot ant starts at (0, 0) and needs to move as quickly as possible in each of the designated directions.

(b) Goal distribution of all 10 tasks for Ant-Goal. The simulated robot ant starts at (0, 0) and needs to navigate to each designated goal represented by dots in the figure.

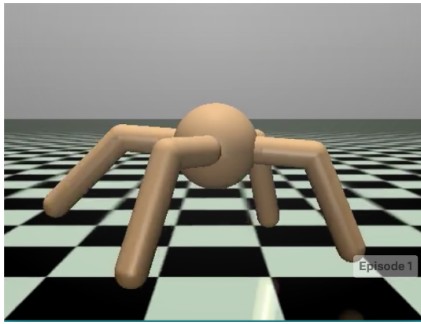

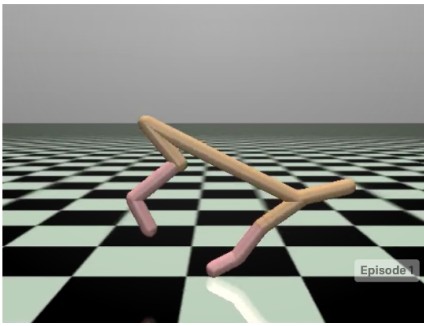

(c) An example of the simulated ant robot. This robot is used for both Ant-Dir and Ant-Goal environments.

(d) An example of the simulated half cheetah robot. This robot is used for the HalfcheetahVel environment. The task for this environment is for the half cheetah robot to reach and maintain a specific speed.

Figure 2: Illustrations of the experiment environment and their task distribution.

2019), there are in total 200,000 samples for Ant-Dir, 300,000 samples for Ant-Goal and 60,000 trajectories for HalfCheetahVel. The state measurements for each task follows exactly the OpenAI gym state. No task-specific information, for example goal location or the target velocity, is added to the state. For Ant-Dir the target directions and goals are sampled from a $120°$ circular arc. Figure 2 illustrates these environments in detail.

## 4.2 EXPERIMENT SETUP

In the experiments, we illustrate the performance of our approaches for both Algorithm 1 and Algorithm 2. For single task BAIL, we use a neural network of two layers with 128 neurons for each layer to first approximate the upper envelope function. As mentioned in the previous section, we select $\beta$ such that the top 25% data for each batch will be selected. Afterwards, we use a three-layers MLP with 128 neurons on each layer to approximate the policy via simple behavior cloning method based on supervised learning. Both networks use ReLu activation function (Nair & Hinton, 2010) and are optimized via Adam (Kingma & Ba, 2014) with learning rate 0.001.

For BAIL+, the upper envelope function is approximated by an MLP of the same structure as for BAIL. To estimate the returns $\hat{G}^t$, we directly use the trained reward models from (Li et al., 2019), which is an ensemble of 10 neural networks. We then follow Equation 5 to estimate the return with the discount factor set to 0.99. Then the transfer selection ratio threshold $\tilde{\alpha}$ is determined so that 2.5% of all the source tasks' data is selected. The imitation learning procedure follows the same as for vanilla BAIL.

For our MBAIL method, the distilled policy $\pi$ follows the same structure as before, i.e. three layers of 128 neurons with ReLu activation function. The inference module is parameterized by a two-

| Improvement (%) | Training samples ratio (%) | 5 | 10 | 20 | 40 | 60 | 80 | 100 |
|---|---|---|---|---|---|---|---|---|
| | BAIL (Single) | -0.12 | 3.18 | 5.79 | 14.82 | 19.87 | 22.33 | 23.21 |
| Ant-Dir | BAIL+ | 0.37 | 3.38 | 6.30 | 14.95 | 20.20 | 21.45 | 22.91 |
| | MBAIL | **5.70** | **5.65** | **5.90** | **16.47** | **18.72** | **23.43** | **23.84** |
| | Contextual BCQ | 0.74 | 2.40 | 4.08 | 11.87 | 14.46 | 18.59 | 19.49 |
| | BAIL (Single) | -76.81 | -51.04 | -50.31 | -50.08 | -49.91 | -50.40 | -49.00 |
| Ant-Goal | BAIL+ | -62.87 | -51.90 | -50.30 | -49.74 | -50.61 | -50.19 | -49.74 |
| | MBAIL | **-50.94** | -49.86 | -50.24 | -47.07 | -47.27 | -46.83 | -47.44 |
| | Contextual BCQ | -72.06 | **-44.09** | **-34.49** | **-30.25** | **-29.30** | **-26.18** | **-27.29** |
| | BAIL (Single) | -22.15 | -22.46 | -22.80 | -22.08 | -20.08 | -17.41 | -13.35 |
| HalfCheetahVel | BAIL+ | -22.09 | -22.72 | -22.52 | -16.05 | -13.33 | -12.87 | -11.22 |
| | MBAIL | **-16.95** | **-16.85** | **-16.76** | **-15.12** | **-12.43** | -12.04 | -11.34 |
| | Contextual BCQ | -20.50 | -20.77 | -20.44 | -16.03 | -12.69 | **-10.69** | **-9.05** |

Table 1: Improvement percentage over baseline single task BAIL.

layered ReLu network with 200 neurons in each layer. We also use Adam to optimize both of these networks with 0.0005 learning rate.

Contextual BCQ have the same components as the vanilla BCQ (Fujimoto et al., 2019), including a Q function, a perturbation network and a variational autoencoder (Kingma & Welling, 2013). For contextual BCQ, we use the exact same model structure as in (Li et al., 2019), where the Q function is estimated via a 9 layers neural network with 1024 neurons for each layer, the perturbation network is parameterized via 8 layers neural network with 1024 neurons and the VAE has 7 layers of the same number of neurons.

### 4.3 EXPERIMENT RESULTS

To evaluate our models, we run the learned policies on each of the training environments for 1,000 steps and compute their returns. We then normalize all the returns for each of the tasks by the average of the Monte Carlo estimated in-batch returns $\bar{G}^t$ for task $t$. The reason for this normalization is that each task, although in the same environment, may differ in reward scaling (e.g., it may be intrinsically easier to walk slowly or reach a nearby goal compared with fast running or distant goals) - note we inherit this property from the public dataset and have not modified the environment code or raw data. Next, we average all the normalized returns to form an evaluation on the current task. In general, for a policy $\pi$, let $r_i^\pi$ denote reward collected at time step $i$ when following the policy $\pi$, we evaluate all the learned policies by Equation :

$$E_t^\pi = \sum_{i=1}^{L} \frac{r_i^\pi}{|\bar{G}^t|} \tag{9}$$

To illustrate the benefit of sample transfer as well as multitask learning, we show the results using different sizes of the training batches. Because the batch for each environment varies in sample size, for the simplicity in the result presentation, we evaluate the models with different percentages of the training batches. We take the first $x\%$ of the batch as the training batch, where $x$ takes the value of $5, 10, 20, 40, 60, 80, 100$, respectively. The averaged normalized returns for different training sample ratios are listed in Table 1 and the learning curves for all environments on training sample ratio 0.1 and 0.8 are presented in Figure 3.

### 4.4 RESULTS ANALYSIS

From Table 1, we can first clearly observe that our BAIL+ and MBAIL outperform the single trained BAIL method for nearly all of the environments. This is especially the case when encountering smaller training sample sizes. This shows that our algorithms, by leveraging information from other tasks, whether it's from sample transfer or policy distillation, have on average improved the policy performance on each task.

More specifically, from Table 1, we can see that BAIL+ consistently outperforms single task BAIL. This result shows that BAIL+, by transferring sample from source tasks to the target task, has improved the performance on the original single task BAIL policy. In addition, as one can see from the

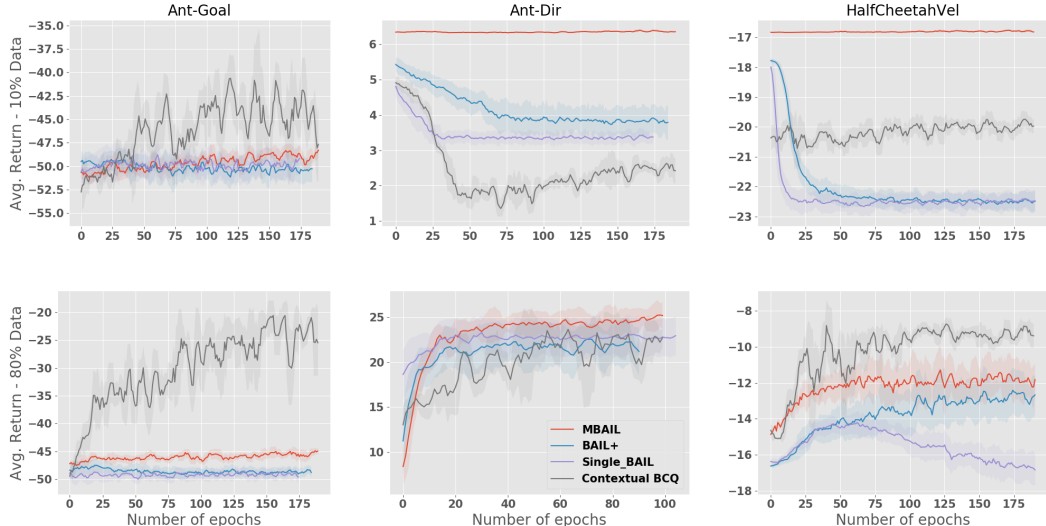

Figure 3: Learning curves for our methods (MBAIL and BAIL+) and the baseline single task BAIL and contextual BCQ. The top row is trained on a batch of sample ratio 0.1, that is 30,000 training examples for Ant-Goal, 20,000 for Ant-Dir and 6,000 for HalfCheetahVel. The bottom row is trained on a batch of sample ratio 0.8, that is 240,000 training examples for Ant-Goal, 160,000 for Ant-Dir and 48,000 for HalfCheetahVel.

learning curve figures 3, when the sample size is too small, namely 20,000 for Ant-Goal and 6,000 for HalfCheetahVel, there is some model degeneration happening when training for longer epochs. We can see that although model degeneration does happen for BAIL+, it can often converge to a better solution. Moreover, in HalfCheetahVel, even with 48,000 examples, this degeneration still happens for BAIL, while for BAIL+ it does not affect its learning. These results further showcase the benefits of BAIL+ compared to BAIL by simply enriching the target batch's dataset.

Another observation is that our MBAIL method consistently outperforms BAIL+ across almost all different sample sizes, as shown in Table 1. This indicates that the policy distillation process of our MBAIL method does further improve the policy's performance on top of the BAIL+ algorithm. Unlike single BAIL or BAIL+, even with low amount of data, as shown in the Figure 3, the distilled policy does not seem to have model degeneration as observed for other methods, indicating a stable performance under small data regime.

One interesting result is that it seems on Ant-Goal environment, none of the BAIL-based methods achieves the desired level of return. This behavior persists across all sample sizes. It appears that BAIL is struggling to obtain a meaningful policy for this particular environment. One potential reason is that the upper envelope function approximated for this environment is not accurate enough, thus resulting in sub-optimal batches to be selected. Further analysis of the failure cases and relative performance of BAIL across conditions is an important question, but one left for future work. Despite the weakness in all BAIL-driven methods for this one environment, MBAIL still outperforms single task BAIL and BAIL+, further indicating that our approach for multi-batch BAIL works as expected.

## 5 CONCLUSION AND FUTURE WORK

Sample efficiency still remains a main obstacle for most reinforcement learning (RL) algorithms be applicable for real-world applications. Batch reinforcement learning has shown to be promising to deal with real-world sequential decision making problems. To further improve the batch RL model's performance with a limited amount of training data points and its performance over multiple tasks, in this work, we propose to tackle these problems with sample transfer and policy distillation. Experiment results show that the proposed methods (BAIL+ and MBAIL) can significantly outperform other baselines when the training data is limited. In the future, we plan to do more evaluation of the proposed methods and investigate how to improve the policy distillation method used in this work.

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
