# OpenReview forum: "Multi-batch Reinforcement Learning via Sample Transfer and Imitation Learning"
_ICLR.cc/2022/Conference — ICLR 2022 Submitted_

### Official Review · Reviewer_ZN2n · 2021-10-31

**Correctness:** 3
**Technical Novelty And Significance:** 2
**Empirical Novelty And Significance:** 2
**Recommendation:** 3
**Confidence:** 5

**Main Review:**

$\textbf{Strengths}$
The problem studied by this paper is interesting, and the motivation is clearly expressed. Overall, the method makes sense to me and achieves improvements when there are very few training samples for each batch.

$\textbf{Weaknesses}$
This paper needs lots of polishment in writing, especially for the second half. Some explanations are needed for the KL term in Equation (6), Equation (7) contains many errors and there are many typos in the paper, e.g., for section 3.2, I think your method is named as multi-batch BAIL not multi-batch BAIL+. More descriptions are needed for each figure.

It looks like most of the components in the proposed algorithms are from previous works, I think the authors should clarify their contributions.

The experimental results are not convincing, with contextual BCQ almost dominating the Ant-Goal game. I also recommend the authors to add more baselines, e.g., CDS [1] (just for example since this one is quite recent, but authors should consider some SOTA batch RL algorithms).

[1] Conservative Data Sharing for Multi-Task Offline Reinforcement Learning. Yu et al.

**Summary Of The Paper:**

This work makes use of transfer learning to deal with the multi-batch reinforcement learning problem. It improves the BAIL algorithm on the multi-batch setting and further extends it to a multi-task model, it empirically demonstrates the advantage in three simulated environments.

**Summary Of The Review:**

This paper proposes two algorithms for the multi-batch RL problem, and experimental results demonstrate their advantage to some extent. However, the writing of this paper needs lots of improvements, the technical contribution is limited, and the final results are not solid enough to support the claims, I feel this work is incomplete and recommend rejection.

---

> ### Author Response · Authors · 2021-11-23
> **Response to Reviewer ZN2n**
>
> Thanks for your comments!
>
> Concerns on the contribution of this work.
>
> We thank the reviewers’ comments on the novelty of the work. Our contribution in this work is to first propose a challenging problem to be solved under multitask batch RL setting, which is to learn one single policy that performs reasonably well across all different tasks with no further interaction with the environment. To solve this problem, we propose the two-stage method, that is, to first learn individual policies, leveraging knowledge from other tasks, then distill these policies into one. For the first stage, we use the sample transfer technique on top of the BAIL algorithm to form our BAIL+ algorithm, extending BAIL to its multitask variant. For the second stage, we modify the method proposed in Li et al., 2019, so that it works with the imitation learned policies obtained by BAIL+.
>
> Concerns on the experiments
>
> We fully agree that it will be better to add more experiments to further analyze the proposed method. We are currently working on adding more experiments, including comparisons with more baseline methods and one additional environment.
>
> Concerns on the writing
>
> Thanks a lot for your detailed comments. We are working on reorganizing the paper and further polishing the paper.

---

> > ### Comment · Reviewer_ZN2n · 2021-11-28
> > **Thanks for the response**
> >
> > Thank you for the response in clarifying the contributions, I think the motivation is clear to me now, but this paper does need a lot of improvements, I'll keep the current score but hope all these suggestions are useful for you.

---

### Official Review · Reviewer_A9qi · 2021-10-31

**Correctness:** 3
**Technical Novelty And Significance:** 1
**Empirical Novelty And Significance:** 2
**Recommendation:** 3
**Confidence:** 3

**Details Of Ethics Concerns:**

This paper does not raise ethics concerns beyond what is normal for batch RL applications.

**Main Review:**

The paper rightfully pointed out that single-batch setting and a large number of interaction samples limited the use cases of offline RL in the real world. The paper provided a viable solution to the multi-task setting with some assumptions to the environment and the data collected. The evolution from BAIL to BAIL+ to MBAIL is clear and the experiments showed clearly that MBAIL is the best out of the three.

There are several aspects that make me hesitant to recommend an acceptance.
* Assuming that there will be new SOTA batch RL algorithms in the future, do the authors have plans to make this work more applicable to non-BAIL algorithms? It is unclear from the paper how easy or hard that would be. If it is hard, then the community will benefit less from this piece of work.
* Section 3.2 is heavily inspired by MBML (Li et al 2019), where the authors made some modifications to make it applicable to an imitation learning based policy. Is it intentional that you do not compare your algorithm with Li et al 2019 in the experiments section?
* The major technical contribution in BAIL+ is that it proposed a way to use the inductive bias that the transition function remains constant for all tasks and the target task appeared in the batch dataset. But with such assumptions, it also limits its use case to a narrow set of dataset and environments. Clear and strong performance can sometimes justify having a narrow focus, but as the experiment result shows, MBAIL does not consistently perform better than contextual BCQ (albeit the latter has privileged info -- namely the identity of the task).


Clarification questions:
* In Figure 3, does the number of epochs mean the same thing for BAIL+ and MBAIL? e.g. for MBAIL I think it is the constant E in algorithm 2. What about BAIL+? It would be great to further clarify.
* in 3.1, it states “Note that we assume all tasks share the same transition function, and the batch is collected via the same policy”. Is “the batch” here referring to all data or the batch of data per task?

Suggestions:
* Would it be possible to list out the assumptions more explicitly? I can see several important assumptions spread throughout the paper and it’s hard to keep track of them. e.g. 1. data batches are labeled with their respective tasks; 2. The tasks only differ by their reward function and they all share the same transition function. 3. The target task is known for BAIL+ and is a part of the batch dataset for both BAIL+ and MBAIL. In the meantime, the introduction made it sound like the algorithm is quite general.
* In the introduction on the second page, you have several sentences that describe how your work differs from previous works and I think the writing there can be improved. E.g. I first read “ In this work, we use sample transfer, ... The proposed algorithm is referred to as BAIL+” and then I read “In this work, ..., we propose to first improve the learning performance on single tasks and then utilize the policy distillation to combine the learned policies into one single policy.” it is confusing because I wasn’t sure whether you use sample transfer, or you use policy distillation, or both. Only later did I realize one refers to BAIL+ and the other refers to MBAIL.

Typos:
* Section 2 “... (Fujimoto et al., 2019) proposed an algorithm to learn policies with soft constrain to lie near the batch, which alleviate the extrapolation problem.” -> “... (Fujimoto et al., 2019) proposed an algorithm to learn policies with soft **constraints** to lie near the batch, which **alleviates** the extrapolation problem.”
* Section 2 “and then learning a control policy based on these selection state-action pairs.”-> “these **selected** state-action pairs.”


**Summary Of The Paper:**

(Note that the rest of the review will use Offline RL and Batch RL interchangeably)

Building on top of BAIL (Chen et al. 2019), this paper provides two algorithms for the multi-task offline RL setting. The BAIL+ algorithm assumes that we know the identity of the target task and that all tasks share the same transition function. It uses sample transfer from non-target task data to boost its performance on the target task. The MBAIL algorithm builds on top of BAIL+ and MBML (Li et al 2019), where it distills the BAIL+ policies for each individual task into one master policy that performs decently on all tasks.
The authors evaluated the proposed methods on the Ant-Dir, Ant-Goal, and HalfCheetahVel environments from Mujoco. The authors compared MBAIL, BAIL+, BAIL, and contextual BCQ. MBAIL consistently performed the best out of the first three, while contextual BCQ performed better on the Ant-Goal env.



**Summary Of The Review:**

Although the paper showed that MBAIL consistently outperforms BAIL under the multi-task setting, I struggle to recommend acceptance because 1. The algorithm is closely tied to one specific offline RL algorithm, which limits its use cases 2. There could be more baseline comparisons in the experiments section 3. The main technical contribution is specific to tasks where they all share the same transition function, which further limits its use cases.

---

> ### Author Response · Authors · 2021-11-23
> **Response to Reviewer A9qi**
>
> Thanks for your detailed comments!
>
> Concerns on using the proposed method on other batch RL algorithms
>
> Our work is mainly focused on BAIL, including extending it to BAIL+ and MBAIL. Thus, how to use the proposed solution for other batch RL algorithms is not well discussed. Thanks for your suggestion. We agree that this work can be more impactful if the proposed solution can be agnostic to the base algorithm.
>
> Concerns on the assumption of the proposed method
>
> As you pointed out, there are two major assumptions for our method. All the data batches are labelled with their respective tasks; all tasks share the same state, action space and transition function while differing in their reward functions. Although the target task terminology is used, we are actually mainly focused on the multi-tasking learning scenario. Therefore the algorithm does not need to know the target task.
>
> Concerns on the writing
>
> Thanks for your detailed comments and suggestions.  We are working on reorganizing the paper and further polishing the writing.
>
> Concerns on the experiments
>
> Thanks for your suggestions. We fully agree with the points you pointed out. We are adding more experiments, including comparisons with more baseline methods and one additional environment.

---

> > ### Comment · Reviewer_A9qi · 2021-11-29
> > **Response to rebuttal**
> >
> > Dear authors,
> >
> > Thank you for the rebuttal and the comments.
> >
> > > Our work is mainly focused on BAIL, including extending it to BAIL+ and MBAIL. Thus, how to use the proposed solution for other batch RL algorithms is not well discussed. Thanks for your suggestion. We agree that this work can be more impactful if the proposed solution can be agnostic to the base algorithm.
> >
> > You do not have to propose a solution in this paper. However, I am arguing that as a part future work discussion, how to use the proposed solution for other batch RL algorithms should be contemplated, or at least mentioned.
> >
> > Regarding the clarity and the experiments, since the authors are still working on the updates, it does not allow me to accept the paper. I hope my comments are helpful. I look forward to seeing either a more general algorithm, or a stronger set of experiment results from the next version of the paper.

---

### Official Review · Reviewer_u4cA · 2021-11-02

**Correctness:** 3
**Technical Novelty And Significance:** 3
**Empirical Novelty And Significance:** 3
**Recommendation:** 5
**Confidence:** 4

**Main Review:**

Pros:

the method proposes a good new baseline method for multitask batch reinforcement learning, which is the main justification for my score

Things to improve:

(1) there is a room for improvement of the experimental analysis (see below)

(2) on the novelty, the method heavily relies upon existing ideas (especially MBAIL which combines policy distillation of (Rusu et al, 2015) and triplet loss (Li et al, 2019)), which, in the reviewer’s opinion, there is nothing wrong about and good methods are often a good combination of existing ideas; however it would be helpful if the authors could state explicitly (for example in the introduction), which new contributions have been made in this paper. This would help readers see the particular novelties of this approach and build upon them.

Improvement of the experimental analysis:

(1)While it is shown that the BAIL+ method improves upon BAIL on a set of proposed tasks, it still would not answer whether it would also improve the distilled policy of MBAIL. To answer this question, is it possible to evaluate an ablation of MBAIL with BAIL instead of BAIL+ as the backbone for distillation?

(2)More insight should be given into why and when it works better. In (Li et al, 2019), for example, they additionally experiment with UMazeGoal-M, HumanoidDir-M and HalfCheetahVel, as well as with WalkerParam, exhibiting different transition functions for  each task.The latter would be extremely interesting in the light of the ablation study proposed in point (1), so it might be possible to see the behaviour of different methods with different transition functions.  Is it possible to give results on any of these tasks?

Minor comments:
(1) Page 3, section BAIL:Best Action Imitation Learning: A minor comment on clarity of writing. For someone browsing through some parts of the section it might look like some parts of BAIL are proposed by the authors (especially given the algorithm below):  "To do this, we estimate a supremum of the optimal value function V ∗, which is referred to as the upper envelope.”

(2) So it might be a good idea to somehow rephrase the parts of the text in this section referencing to ‘we' to improve clarity (the reviewer understands that it is the background section so it describes the existing work, but nevertheless).

==
UPDATE: as the comments (mine and other reviewers) have not been addressed yet, there are still open questions which unfortunately do not allow recommending acceptance as things stand at the moment.

**Summary Of The Paper:**

The paper describes multiple task batch reinforcement learning method, building upon a well-known single task batch reinforcement method BAIL. The method proposes two separate improvements, namely BAIL+, introducing the idea of leveraging data across tasks with the same transition function, and MBAIL, policy distillation approach for multiple batch learning.

**Summary Of The Review:**

Recommending acceptance subject to convincing experimental analysis improvement and improved description of the novelty of hte approach

---

> ### Author Response · Authors · 2021-11-23
> **Response to Reviewer u4cA**
>
> Thanks a lot for your detailed comments and suggestions. We fully agree with the points you mentioned. We are working on implementing more experiments and will add more discussions on when the proposed method will work and why it can work.

---

> > ### Comment · Reviewer_u4cA · 2021-11-28
> > **Rebuttal score update**
> >
> > As the comments (mine and other reviewers) have not been addressed yet in the revised paper, there are still open questions which unfortunately do not allow recommending acceptance as things stand at the moment. However, I hope the reviewers' suggestions could help improve the paper.

---

### Official Review · Reviewer_CMTJ · 2021-11-02

**Correctness:** 2
**Technical Novelty And Significance:** 2
**Empirical Novelty And Significance:** 2
**Recommendation:** 3
**Confidence:** 4

**Main Review:**

=======strengths=======
- The first stage of improving on each task before distilling seems to be an interesting novelty.
- The problem of multi-task batch RL is a direction that is not adequately studied.

=======weaknesses=======
- the contribution of the paper seems a bit lacking, a large portion of BAIL+ and MBAIL are based on the multi-task batch RL methods described in Li et al., 2019, the first stage in the proposed 2-stage method is probably novel, but I'm not sure if the rest of the technical content are significantly novel.
- lack of comparison to previous methods, in particularly, while a majority of the work is based on the MBML algorithm and results from Li et al., 2019, the proposed method is not compared to MBML, and the figures use a "normalized" yaxis, making them look entirely different from what is in the Li et al. paper, thus it becomes impossible to compare the two methods.
- lack of reasoning on why the proposed method would work better than other previous methods. This is partly due to lack of comparison to previous methods, but the authors also provided very little theoretical/empirical analysis on why the proposed method would work well. Very little insight is provided. At one point the authors mention the problem of performance drop after training BAIL for a while. This is a very interesting problem but it is not investigated further in the paper.
- Why is BAIL a good base algorithm? What are the advantages of using it together with your 2-stage scheme, authors should consider refine their writing and emphasize the unique advantages of such a design and give better insight on this design choice. For example, perhaps BAIL is simpler and faster than other methods? If that's the case, provide a comparison on computation complexity and provide discussions.
- the paper is lacking in clarity of writing, a number of things are unclear, a large num of typos, things are unclear. Authors should go over the paper carefully and fix all the small issues.
- page 5 top "we assume all tasks share the same transition function, and the batch is collected via the same policy", really unclear what "the same policy" means? You use the same policy to collect data in different tasks?
- page 8 Table 1, really not sure what is the difference between "BAIL (single)" and the "baseline single task BAIL", I assume these are two different things? What is more confusing is that the numbers in the table are mostly negative.
- Table 1, the difference between some of the reported performance numbers are quite small, but I don't see where you report the number of seed used to get these performance and there is no std reported. How statistically significant are these comparisons?
- page 4 bottom "by assuming the correct estimation of..." why is it that such an assumption is realistic?


**Summary Of The Paper:**

This paper studies the problem of multi-task offline RL and propose 2 methods, BAIL+ and MBAIL, each is an algorithmic variant based on the BAIL algorithm. The overall goal is to improve performance on multiple tasks. This setting concerns with a number of tasks coming from a MDP distribution where transition function is the same and reward function is different.

The authors propose a 2-stage method: first use the BAIL+ algorithm to boost the performance of a particular task with data from other tasks. Then, distillation is used to distill policies from these tasks into one policy.

**Summary Of The Review:**

- A major issue here is lack of consistency and adequate comparison with prior work, especially for Li et al., 2019, where the authors proposed and studied reward relabeling, multi-task distillation and use the triplet loss. It becomes a bit unclear how significant the technical contributions. This also affects the contribution on empirical results: it is hard to evaluate their significance without a proper comparison to prior methods. I recommend the authors try to design other analysis or ablations to showcase why the proposed design is a good choice. And to further investigate some of the interesting observations such as the performance drop issue after training BAIL for a long time.
- Another major issue is the clarify of the paper. Authors should go through the paper slowly and refine the writing, this is more than just minor issues on grammar and typos, the writing in the paper should be very clear and make sense, any claims should be made with supportive evidence, etc.
- The paper proposes some interesting ideas, and the multi-task batch RL setting is relatively not well explored, but due to the large number of issues, it's hard to believe the paper is ready for publication in its current state.

---

> ### Author Response · Authors · 2021-11-23
> **Response to reviewer CMTJ**
>
> Concern on the contribution.
>
> We thank the reviewers’ comments on the novelty of the work. Our contribution in this work is to first propose a challenging problem to be solved under multitask batch RL setting, which is to learn one single policy that performs reasonably well across all different tasks with no further interaction with the environment. To solve this problem, we propose the two-stage method, that is, to first learn individual policies, leveraging knowledge from other tasks, then distill these policies into one. For the first stage, we use the sample transfer technique on top of the BAIL algorithm to form our BAIL+ algorithm, extending BAIL to its multitask variant. For the second stage, we modify the method proposed in [1], so that it works with the imitation learned policies obtained by BAIL+.
>
> Reason for using BAIL as the base algorithm.
>
> We use BAIL mainly for its great sample efficiency. This advantage is especially helpful under the batch setting as data is very scarce.
>
> Lack of reasoning why it works and concerns about the writing.
>
> We thank the reviewer for the detailed comments. We will reorganize the paper, further improve this paper's writing, and add additional discussions to explain why it works.
>
> Concerns on the writing
>
> Thanks again for your detailed comments. We are working on reorganizing the paper and further polishing the paper.
>
> [1] Li J, Vuong Q, Liu S, et al. Multi-task batch reinforcement learning with metric learning[J]. arXiv preprint arXiv:1909.11373, 2019.

---

> > ### Comment · Reviewer_CMTJ · 2021-11-29
> > **Response to rebuttal**
> >
> > Thank you for the response! I think the main idea of the paper is interesting, and I agree that using BAIL here can have some advantages. I will keep my evaluation since to address all reviewer concerns will take significant modifications to the paper and would require another review evaluation. But I hope the comments from me and other reviewers can help the authors to greatly improve the paper.
> >
> > Perhaps the most important improvements would come from:
> > - refine the overall writing, and especially try to improve the clarity
> > - provide more analysis and insights on the proposed method and experiments
> > - state more clearly how your work is related to previous work and what are your novelty and contributions, and when necessary include more important baselines
> > - emphasize the advantage of your proposed method

---

### Decision · Program_Chairs · 2022-01-20

**Decision:**

Reject

**Comment:**

The main identified issues were the limited contribution and use cases, poor writing and missing baseline comparisons and more needed experiments. These issues were not addressed satisfactorily by the rebuttal and hence, I believe the paper should be revised by the authors and undergo another review process at another conference. I therefore recommend rejection.